# Hemodynamic Effects of Permanent His Bundle Pacing Compared to Right Ventricular Pacing Assessed by Two-Dimensional Speckle-Tracking Echocardiography

**DOI:** 10.3390/ijerph182111721

**Published:** 2021-11-08

**Authors:** Jedrzej Michalik, Alicja Dabrowska-Kugacka, Katarzyna Kosmalska, Roman Moroz, Adrian Kot, Ewa Lewicka, Marek Szolkiewicz

**Affiliations:** 1Kashubian Center for Heart and Vascular Diseases, Department of Cardiology and Interventional Angiology, Pomeranian Hospitals, 84-200 Wejherowo, Poland; jedri1616@gmail.com (J.M.); romanmoroz@wp.pl (R.M.); aadriankot@gmail.com (A.K.); 2Department of Cardiology and Electrotherapy, Medical University of Gdansk, 80-214 Gdansk, Poland; alidab@gumed.edu.pl (A.D.-K.); ewa.lewicka@gumed.edu.pl (E.L.); 3Department of Cardiology, Pomeranian Hospitals, 81-348 Gdynia, Poland; katarzyn5@wp.pl

**Keywords:** His bundle pacing, ventricular synchrony, 2D speckle-tracking echocardiography, global longitudinal strain, left atrial volume

## Abstract

We compared the effects of right ventricular (RVP; *n* = 26) and His bundle (HBP; *n* = 24) pacing in patients with atrioventricular conduction disorders and preserved LVEF. Postoperatively (1D), and after six months (6M), the patients underwent global longitudinal strain (GLS) and peak systolic dispersion (PSD) evaluation with 2D speckle-tracking echocardiography, assessment of left atrial volume index (LAVI) and QRS duration (QRSd), and sensing/pacing parameter testing. The RVP threshold was lower than the HBP threshold at 1D (0.65 ± 0.13 vs. 1.05 ± 0.20 V, *p* < 0.001), and then it remained stable, while the HBP threshold increased at 6M (1.05 ± 0.20 vs. 1.31 ± 0.30 V, *p* < 0.001). The RVP R-wave was higher than the HBP R-wave at 1D (11.52 ± 2.99 vs. 4.82 ± 1.41 mV, *p* < 0.001). The RVP R-wave also remained stable, while the HBP R-wave decreased at 6M (4.82 ± 1.41 vs. 4.50 ± 1.09 mV, *p* < 0.02). RVP QRSd was longer than HBP QRSd at 6M (145.0 ± 11.1 vs. 112.3 ± 9.3 ms, *p* < 0.001). The absolute value of RVP GLS decreased at 6M (16.32 ± 2.57 vs. 14.03 ± 3.78%, *p* < 0.001), and HBP GLS remained stable. Simultaneously, RVP PSD increased (72.53 ± 24.15 vs. 88.33 ± 30.51 ms, *p* < 0.001) and HBP PSD decreased (96.28 ± 33.99 vs. 84.95 ± 28.98 ms, *p* < 0.001) after 6 months. RVP LAVI increased (26.73 ± 5.7 vs. 28.40 ± 6.4 mL/m^2^, *p* < 0.05), while HBP LAVI decreased at 6M (30.03 ± 7.8 vs. 28.73 ± 8.7 mL/m^2^, *p* < 0.01). These results confirm that HBP does not disrupt ventricular synchrony and provides advantages over RVP.

## 1. Introduction

Cardiac electrotherapy is a rapidly developing field of medicine. Modern biomedical technology provides opportunities to create devices that are small, durable and safe, with complex delivery systems or even leadless. Stimulation methods have also changed with the development of new technologies. Previously, we sought effective methods of stimulation, but we now seek methods that are both effective and physiological.

Right ventricular pacing (RVP) is common and easy to use. It requires no extraordinary surgical skills, allows practitioners to obtain adequate sensing and pacing parameters and causes minimal periprocedural complications. Unfortunately, it can cause both electrical and mechanical ventricular dyssynchrony, which diminishes the left ventricular (LV) function and may result in more frequent atrial fibrillation, heart failure and even death [1,2]. 

A significant milestone in electrotherapy was the introduction of biventricular stimulation, which partially restores the synchrony of contractions in both ventricles and improves LV systolic function—benefits that are especially noticeable in patients with reduced LV ejection fraction (LVEF) and left bundle branch block. However, this method requires an additional pacing lead and is not suitable for all patients [3].

His bundle pacing (HBP), first described in 2000 by Deshmukh et al., appears to be the most physiological (and, therefore, optimal) form of cardiac stimulation. By pacing the native His-Purkinje system, HBP provides direct and synchronous stimulation of both ventricles [4,5]. Recent studies have shown that, although this is not a flawless method (it is technically difficult, generates low R-wave amplitudes, has a high rate of electrode dislocation and involves a high and often unstable stimulation threshold), it is feasible, produces a narrow QRS duration and prevents the development of pacing-induced cardiomyopathy [6]. Notably, in some cases, HBP efficiently corrects pre-existing bundle branch block, shortening the QRS duration and restoring the physiological synchrony of contractions [7].

Two-dimensional speckle-tracking echocardiography (2D STE) enables quantitative analysis of the degree of deformation in myocardial segments, as well as the scale of contraction dyssynchrony. Previous studies have shown that global longitudinal strain (GLS) and peak systolic dispersion (PSD) enable a more objective, accurate and reproducible assessment of myocardial dysfunction compared with tissue doppler imaging (TDI) or measurement of LVEF [8], which, so far, have been mainly used to assess the hemodynamic effects of HBP [9,10]. Left atrial volume is a recognised indicator of LV diastolic dysfunction, and it is a good predictor of atrial fibrillation [11]. The aim of this study was to analyse the safety and hemodynamic effects of chronic HBP compared with RVP using 2D STE.

## 2. Materials and Methods

This single-center, prospective, observational study was performed in Pomeranian hospitals. The participants (*n* = 50) were consecutive patients scheduled for permanent pacing therapy according to the current guidelines. The protocol was approved by the local ethics committee (KB-35/21) and all participants provided written informed consent. Patients at least 18 years of age, with a need for frequent (≥70%) or continuous ventricular pacing, and with normal left ventricular systolic function (EF ≥ 50%) were considered for enrollment in the study. Exclusion criteria were chronic congestive heart failure, acute coronary syndrome, cardiomyopathy, advanced kidney or liver disease and any infectious disease. Patients with previously implanted cardiac pacing devices were also excluded. 

Patients were divided into two groups: Group I (*n* = 26) underwent pacemaker implantation with RVP; Group II (*n* = 24) underwent pacemaker implantation with HBP. Right ventricular leads were implanted in standard mode. If required, the atrial lead was typically placed in the right atrial appendage. His bundle leads (Select Secure 3830, 69 cm, Medtronic Inc., Fridley, MN, USA) were placed with the appropriate delivery sheath (C315HIS, Medtronic Inc., Fridley, MN, USA) using an electrophysiology system for His potential mapping. Eight patients in Group I were diagnosed with permanent atrial fibrillation and high-degree atrio-ventricular block (AVB), and 18 were diagnosed with sinus rhythm and AVB. Thirteen patients in Group II were diagnosed with permanent atrial fibrillation and high-degree AVB, and 11 were diagnosed with sinus rhythm and AVB. 

Pacing threshold and R-wave amplitude were recorded using the Medtronic system analyser (model 2090) immediately after the procedure (1D) and after six months (6M). QRS complex duration (QRSd) was obtained via electrophysiological equipment with electronic calipers at a sweep speed of 100 mm/s. QRSd was measured in lead V6 from the onset of intrinsic/paced QRS to the end of the QRS complex, before pacemaker implantation (0D), immediately after the procedure (1D) and at the 6-month follow-up visit (6M). 

All patients underwent transthoracic echocardiography (VIVID S70 with M5Sc transducer, GE Healthcare System, Chicago, Il, USA) before, immediately after implantation (1D) and six months later (6M). This procedure was performed by one experienced sonographer (JM). Only echocardiograms of good quality and with frame rates between 40 and 80 frames per second were included for analysis. Left atrial volume index (LAVI) was calculated by dividing the endsystolic left atrial volume (measured with the area-length method) by the body surface area of patients. Estimation of global longitudinal peak systolic strain and peak systolic dispersion by 2D speckle tracking was performed in standard apical two-, three- and four-chamber views and calculated automatically (automated function imaging) offline using GE EchoPAC software (PC version 201). Moreover, AV delay was programmed postoperatively using echocardiography in order to provide the longest left ventricular diastolic filling time without atrial wave truncation, if applicable, separately for the sensed and the paced mode [12].

Statistical analysis was performed using IBM SPSS Statistics 18 software (SPSS Inc., Chicago, IL, USA) and Microsoft Excel 2007 software. Continuous data are expressed as mean ±SD, and the statistical significance of differences between the groups was assessed using the Student’s *t*-test (or the Mann–Whitney test if the data were not normally distributed). Categorical data are expressed as percentages, and the statistical significance of differences between the groups was assessed using the χ^2^ test. All *p* values under 0.05 were considered significant.

## 3. Results

The study cohort consisted of 50 consecutive patients (28 men and 22 women), randomly assigned to one of the two pacing sites. HBP was attempted in 26 patients; however, we failed in two cases, due to an unmappable His signal (1) and an unacceptably high His pacing threshold, respectively (1; overall success rate: 92%). In all, we included 24 patients with HBP (13 men and 11 women; 73.0 ± 14.4 years) and 26 with RVP (15 men and 11 women; 77.2 ± 6.7 years). All patients had high-degree atrioventricular conduction disorder, so we anticipated high rates of ventricular pacing (>70%). All the procedures were denovo implantations. Patient characteristics are presented in Table 1. There were no differences in demographic parameters, rates of comorbidities or indication for cardiac pacing between the groups.

At baseline (0D), RVP QRSd was shorter than HBP QRSd (108.1 ± 15.6 vs. 121.7 ± 23.4 ms, *p* < 0.05), but it significantly increased immediately after pacemaker implantation (108.1 ± 15.6 vs. 143.4 ± 9.2 ms, *p* < 0.001) and over the following six months (108.1 ± 15.6 vs. 145.0 ± 11.1 ms, *p* < 0.001). HBP QRSd did not significantly change during this time (121.7 ± 23.4 vs. 112.3 ± 9.3 ms, n.s.). In the 6-month follow-up visit, RVP QRSd was noticeably longer than HBP QRSd (145.0 ± 11.1 vs. 112.3 ± 9.3 ms, *p* < 0.001). These data are presented in Figure 1. The 12-lead ECG data of selected pacing are presented in Figure 2.

There were significant differences in pacing threshold and R-wave amplitude between the RVP and HBP groups (Table 2). Immediately after pacemaker implantation (1D), the pacing threshold in the RVP group was lower than in the HBP group (0.65 ± 0.13 vs. 1.05 ± 0.20 V, *p* < 0.001); then, it remained stable until the 6-month follow-up visit. Meanwhile, the pacing threshold in the HBP group significantly increased (1.05 ± 0.20 vs. 1.31 ± 0.30 V, *p* < 0.001). The R-wave amplitude measured at 1D was higher in the RVP group than in the HBV group (11.52 ± 2.99 vs. 4.82 ± 1.41 mV, *p* < 0.001). It also remained stable until the 6-month follow-up visit, while the R-wave amplitude in the HBP group significantly decreased (4.82 ± 1.41 vs. 4.50 ± 1.09 mV, *p* < 0.02).

Postoperatively (1D), the absolute value of global longitudinal strain (GLS) was greater in the RVP group than in the HBP group (Table 2); however, this difference fell slightly short of significance (16.32 ± 2.57 vs. 14.85 ± 2.52%, *p* = 0.051). After six months, GLS significantly decreased in the RVP group (16.32 ± 2.57 vs. 14.03 ± 3.78%, *p* < 0.001) and remained stable in the HBP group. No significant difference was found in GLS between the two groups six months after pacemaker implantation (14.03 ± 3.78 vs. 14.98 ± 1.96%, n.s.). Additionally, we analysed whether there were any differences in GLS between patients with sinus rhythm and AVB (SR+AVB) vs. patients with atrial fibrillation and AVB (AF+AVB). Postoperatively (1D), the absolute value of GLS in the RVP group was greater in the SR+AVB patients than in the patients with AF+AVB (16.93 ± 2.5 vs. 14.7 ± 2.1%, *p* < 0.05), and this difference was maintained after 6 months of follow-up (15.00 ± 3.5 vs. 11.68 ± 3.3%, *p* < 0.05). There were no significant differences in the absolute value of GLS SR+AVB vs. AF+AVB in the HBP group at 1D or 6M. It is worth noting that the absolute value of GLS in the RVP group decreased over 6 months in both SR+AVB and AF+AVB patients (16.93 ± 2.5 vs. 15.00 ± 3.5%, *p* < 0.005; 14.70 ± 2.1 vs. 11.68 ± 3.3%, *p* < 0.005; respectively), while it remained stable in both analysed groups of patients if the His bundle was paced. The data are presented in Table 3.

Immediately after implantation (1D), peak systolic dispersion (PSD) was greater in the HBP group than in the RVP group (96.28 ± 33.99 vs. 72.53 ± 24.15 ms, *p* < 0.01). After six months, PSD significantly increased in the RVP group (72.53 ± 24.15 vs. 88.33 ± 30.51 ms, *p* < 0.001) and significantly decreased in the HBP group (96.28 ± 33.99 vs. 84.95 ± 28.98 ms, *p* < 0.001). No significant difference was found in PSD between the two groups at the six-month follow-up visit (88.33 ± 30.51 vs. 84.95 ± 28.98 ms, n.s.). The data are presented in Table 2. GLS and PSD bull’s eye diagrams are presented in Figure 3.

The left atrial volume index (LAVI) was also measured (Table 2). There was no difference in LAVI between the two groups postoperatively (26.73 ± 5.7 vs. 30.03 ± 7.8 mL/m^2^, n.s.). Over the next six months, LAVI in the RVP group significantly increased (26.73 ± 5.7 vs. 28.40 ± 6.4 mL/m^2^, *p* < 0.05), while it significantly decreased in the HBP group (30.03 ± 7.8 vs. 28.73 ± 8.7 mL/m^2^, *p* < 0.01). However, there was still no difference between the groups at the 6-month follow-up visit (28.40 ± 6.4 vs. 28.73 ± 8.7 mL/m^2^, n.s.).

## 4. Discussion

HBP is a natural step in the development of cardiac electrotherapy that enables the physiological stimulation of the heart. Recent research has demonstrated that HBP is justified both theoretically and practically. Physiological stimulation of the heart is supposed to improve electrocardiographic and echocardiographic parameters and provide beneficial effects in terms of quality of life, morbidity and even mortality. We demonstrated that HBP provides significant benefits over the commonly used RVP in terms of its electrocardiographic and echocardiographic parameters, which can be assumed to improve quality of life and reduce morbidity and mortality over the long term.

HBP is assumed to be more difficult to install than RVP, and, furthermore, HBP’s achieved pacing/sensing parameters are worse than RVP’s. Our results confirm these assumptions. The stimulation threshold in the HBP group immediately after implantation was fully acceptable, but it was higher than the RVP group’s pacing threshold. Moreover, this threshold increased slightly over the six-month follow-up period, while it remained stable in the RVP group. The R-wave in the RVP group was also high and stable. In the HBP group, the R-wave was significantly lower immediately after the procedure, and it decreased further after six months. Similar differences were noticed in previous studies [13,14]. Group differences in the observed values do not, however, arouse any practical concern. While the above data are more favourable in the RVP group, which could be better in terms of pacemaker battery life, the data from both groups fall within ranges that ensure safe and effective pacing. However, the expected clinical benefits should outweigh the risk of greater energy loss and presumably faster battery consumption.

Electromechanical left ventricular systolic dyssynchrony causes most of the negative consequences associated with RVP, including pacing-induced cardiomyopathy [15]. This is not a common phenomenon in cardiac electrotherapy, but it is one of the most important reasons that researchers have focused more on physiological stimulation. QRSd is a good indicator of electrical synchrony. Our results show that, unlike RVP, HBP does not disrupt QRSd, triggering a simultaneous depolarization of the ventricles. QRSd in the RVP group significantly increased immediately after the procedure and (although less significantly) over the following six months. On the other hand, QRSd in the HBP group did not change significantly after the procedure, remaining relatively stable throughout the observation period. The stimulation mode influenced not only electrical but also mechanical synchrony, as evidenced by the peak systolic dispersion (PSD) measurements obtained using 2D STE. PSD adequately reflects the degree of mechanical dyssynchrony accompanying electrotherapy [16] and is used to assess left ventricular dysfunction in other pathologies as well [17,18]. Our PSD results are consistent with our findings for QRSd. During the six months of follow-up, PSD in the RVP group significantly increased, along with the prolonged QRSd, indicating electromechanical dyssynchrony in these patients. On the other hand, PSD in the HBP group significantly decreased. Similar conclusions have been presented in previous studies. Tang et al. showed a significantly greater PSD in the RVP group than in the HBP group as early as one week after the procedure [14]. Sun et al. observed significantly greater PSD values in the RVP group compared with the left bundle branch pacing (LBBP) group [19]. Bednarek et al. noted higher PSD values in patients with nonselective LBBP compared with patients with selective LBBP [16]. These studies support the hypothesis that the more physiological the stimulation, the lower the PSD values (thus, the lower the mechanical dyssynchrony).

Global longitudinal strain (GLS) is currently perceived as a highly valuable tool for assessing myocardial contractility disorders, much more sensitive than LVEF [8,20]. It is a good predictor of adverse events in patients with heart failure [21]. Studies show that the absolute value of GLS is significantly reduced when RVP is used [22]; GLS helps to predict the development of pacing-induced ventricular dysfunction [23]. Prakash et al. showed that GLS did not change during the six-month follow-up in patients with HBP, regardless of whether HBP was selective or nonselective [24]. In a study by Fehske et al., GLS in the RVP group was significantly reduced, while GLS in the HBV group did not change [25]. Our study corroborates these findings. GLS decreased significantly in the RVP group and did not change in the HBP group. There was no significant difference between the groups at the 6-month follow-up visit. We assume that extending the observation period would allow us to obtain a statistically significant difference. The observed effects were similar in patients with both sinus rhythm and permanent atrial fibrillation.

RVP adversely affects the structure and function of the left atrium [26]. In this study, we compared changes in LAVI, which reflect LV diastolic dysfunction. Increased LAVI is also a risk factor for the development of atrial fibrillation [11,27]. LAVI increased significantly in the RVP group and decreased in the HBP group. Pastore et al. also noted that RVP resulted in increased left atrial volume parameters compared with HBP [28]. This is another important feature of the hemodynamic disruptions accompanying RVP—one that can be avoided by changing the method of stimulation. Further studies are necessary to determine whether this intervention leads to a reduction in the incidence of clinical atrial fibrillation.

## 5. Limitations of the Study

This was a prospective, observational study with all the limitations characteristic for single-center studies. The number of patients and follow-up duration were limited and therefore the results should be interpreted with caution. Nevertheless, we demonstrated that HBP, when compared to RVP, maintains electrical and mechanical ventricular synchrony, preventing heart muscle systolic and diastolic function disorders. A larger, randomized, multicenter trial comparing RVP to HBP, with long-term follow-up, is necessary to confirm our findings. 

## 6. Conclusions

The present study confirms the feasibility of HBP, with a high success rate (92%). This method provides adequate pacing and sensing parameters. Above all, unlike RVP, HBP does not disrupt electrical and mechanical ventricular synchrony, which can prevent the remodeling of the heart muscle that leads to systolic and diastolic dysfunction disorders. Thus, HBP maintains the hemodynamic balance, the lack of which leads to arrhythmias, heart failure and an increase in mortality. It is expected that, in the coming years, HBP/LBBP will become a commonly used method of cardiac pacing.

## Figures and Tables

**Figure 1 ijerph-18-11721-f001:**
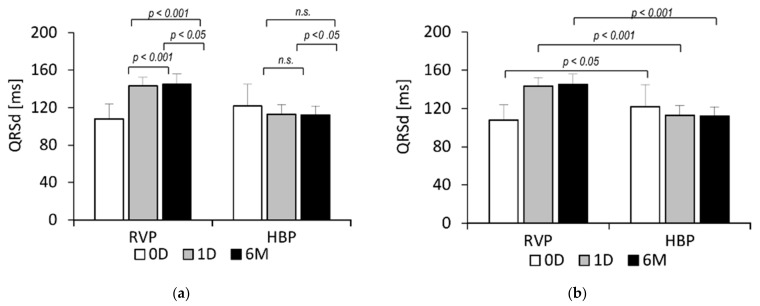
QRS complex duration (QRSd) in patients with right ventricular pacing (RVP) and His bundle pacing (HBP), at baseline (0D), immediately after pacemaker implantation (1D) and 6 months later (6M). Statistical analysis presented: (**a**) within and (**b**) between the studied groups of patients.

**Figure 2 ijerph-18-11721-f002:**
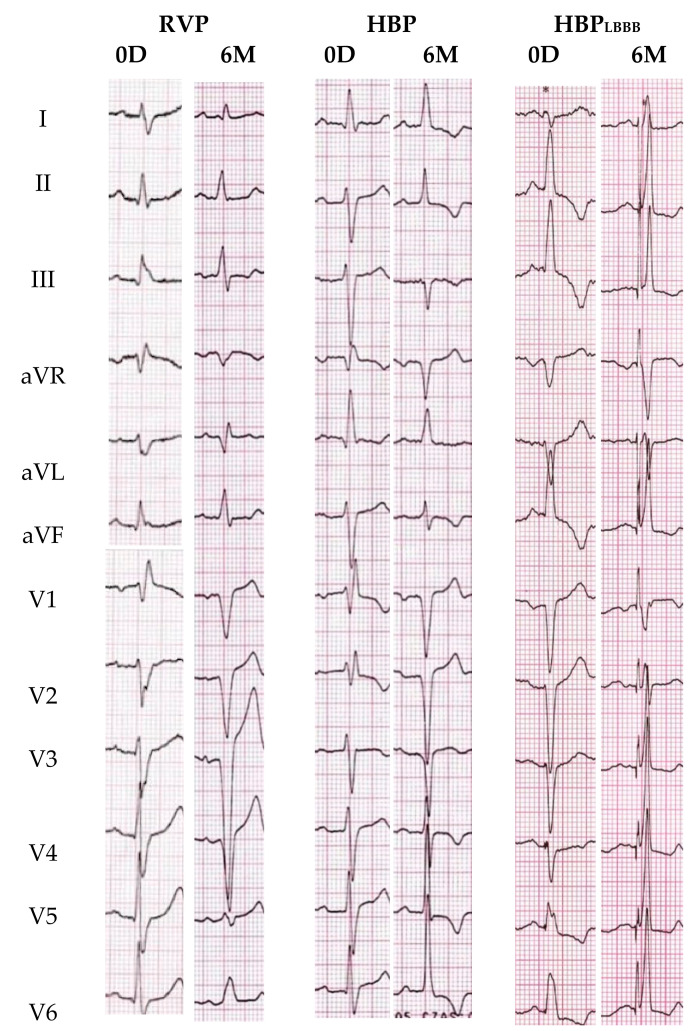
Comparison of 12-lead ECG recorded at a sweep speed of 25 mm/s before (0D) and 6 months after (6M) pacemaker implantation in selected patients treated with right ventricular pacing (RVP), His bundle pacing (HBP) and His bundle pacing in patients with pre-existing LBBB (HBP_LBBB_).

**Figure 3 ijerph-18-11721-f003:**
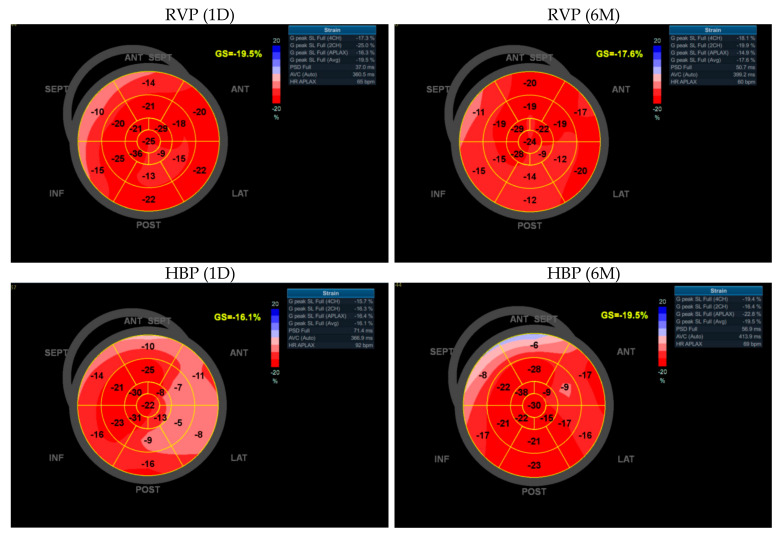
Echocardiographic assessment of left ventricular function using 2D speckle-tracking technique: bull’s eye diagram of global longitudinal strain and peak systolic dispersion of selected patients with right ventricular pacing (RVP) and His bundle pacing (HBP) obtained immediately after pacemaker implantation (1D) and at 6-month follow-up visit (6M).

**Table 1 ijerph-18-11721-t001:** Clinical characteristics of patients with right ventricular pacing (RVP) and His bundle pacing (HBP).

	RVP (*n* = 26)	HBP (*n* = 24)	*p*-Value
Demographics			
Age (years)	77.2 ± 6.7	73.0 ± 14.4	n.s.
Sex (male, n/%)	15/58	13/54	n.s.
Indication for Pacing			
Sick Sinus Syndrome (n/%)	0/0	0/0	n.s.
Sinus Rhythm with AVB (n/%)	18/69	11/46	n.s.
Permanent Atrial Fibrillation with High-Degree AVB (n/%)	8/31	13/54	n.s.
Comorbidities			
Hypertension (n/%)	17/65	16/67	n.s.
Diabetes (n/%)	11/42	8/33	n.s.
Coronary Artery Disease (n/%)	8/31	6/25	n.s.
Chronic Kidney Disease (n/%)	8/31	5/21	n.s.
Atrial Fibrillation (n/%)	10/38	17/71	n.s.
LBBB/RBBB (n/%)	1/4	6/25	n.s.

**Table 2 ijerph-18-11721-t002:** Comparison of selected variables between (and within) RVP and HBP groups of patients immediately after pacemaker implantation (1D) and at 6-month follow-up visit (6M). Abbreviations: RVP—right ventricular pacing; HBV—His bundle pacing; GLS—global longitudinal strain; PSD—peak systolic dispersion; LAVI—left atrial volume index.

	RVP_1D_	HBP_1D_	*p*- Value	RVP_1D_	RVP_6M_	*p*- Value	HBP_1D_	HBP_6M_	*p*- Value	RVP_6M_	HBP_6M_	*p*- Value
Threshold (V)	0.65 ± 0.1	1.05 ± 0.2	<0.001	0.65 ± 0.1	0.66 ± 0.2	n.s.	1.05 ± 0.2	1.31 ± 0.3	<0.001	0.66 ± 0.2	1.31 ± 0.3	<0.001
R-wave (mV)	11.52 ± 3.0	4.82 ± 1.4	<0.001	11.52 ± 3.0	11.41 ± 2.3	n.s.	4.82 ± 1.4	4.50 ± 1.1	<0.02	11.41 ± 2.3	4.50 ± 1.1	<0.001
GLS (-%)	16.32 ± 2.6	14.85 ± 2.5	=0.051	16.32 ± 2.6	14.03 ± 3.8	<0.001	14.85 ± 2.5	14.98 ± 2.0	n.s.	14.03 ± 3.8	14.98 ± 2.0	n.s.
PSD (ms)	72.53 ± 24.2	96.28 ± 34.0	<0.01	72.53 ± 24.2	88.33 ± 30.5	<0.001	96.28 ± 34.0	84.95 ± 29.0	<0.001	88.33 ± 30.5	84.95 ± 29.0	n.s.
LAVI (mL/m^2^)	26.73 ± 5.7	30.03 ± 7.8	n.s.	26.73 ± 5.7	28.40 ± 6.4	<0.05	30.03 ± 7.8	28.73 ± 8.7	<0.01	28.40 ± 6.4	28.73 ± 8.7	n.s.

**Table 3 ijerph-18-11721-t003:** Comparison of global longitudinal strain (GLS) between patients with sinus rhythm and atrioventricular block (SR+AVB) vs. patients with atrial fibrillation and atrioventricular block (AF+AVB) immediately after pacemaker implantation (1D) and at 6-month follow-up visit (6M). Abbreviations: RVP—right ventricular pacing; HBV—His bundle pacing.

	1D	6M	*p*-Value
RVP (SR+AVB)	16.93 ± 2.5	15.00 ± 3.5	<0.005
RVP (AF+AVB)	14.70 ± 2.1	11.68 ± 3.3	<0.005
*p*-value	<0.05	<0.05	
HBP (SR+AVB)	14.37 ± 2.9	15.20 ± 2.1	n.s.
HBP (AF+AVB)	15.76 ± 2.3	15.18 ± 1.9	n.s.
*p*-value	n.s.	n.s.	

## Data Availability

The data supporting the findings of the study are available from Jedrzej Michalik (jedri33@gmail.com) on reasonable request.

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
