# Peer review of "Hemodynamic Effects of Permanent His Bundle Pacing Compared to Right Ventricular Pacing Assessed by Two-Dimensional Speckle-Tracking Echocardiography"

_ijerph, 2021, doi:10.3390/ijerph182111721_

Round 1

Reviewer 1 Report

Thank you for providing me the opportunity to review the paper entitled "Hemodynamic effects of permanent His bundle pacing compared to right ventricular pacing assessed by two dimensional speckle-tracking echocardiography” in International Journal of Environment Research and Public health

General comments:

The study explored to compare the effects of right ventricular (RVP; n=26) and His bundle (HBP; n=24) pacing in patients with atrioventricular conduction disorders and preserved LVEF. Postoperatively (1D), and after six months (6M), the patients underwent global longitudinal strain (GLS) and peak systolic dispersion (PSD) evaluation with 2D speckle-tracking echocardiography, assessment of maximum left atrial volume (LAVmax) and QRS duration (QRSd), and sensing/pacing parameters testing. The authors proposed that the present study confirms the feasibility of HBP, with a high success rate (92%). This method provides adequate pacing and sensing parameters. Above all, unlike RVP, HBP does not disrupt electrical and mechanical ventricular synchrony, which can prevent the remodelling of the heart muscle and its systolic and diastolic function disorders. Thus, HBP maintains haemodynamic balance, the lack of which leads to arrhythmias, heart failure and an increase in mortality. It is a well-written original article, and the findings are well presented. I have some comments.

Major:

  1. In this study, how many patients had single lead or dual leads pacemaker? Because as you know, AV dyssynchrony may affect 2D speckled strain. Especially longitudinal strain.
  2. In page 3, table 1, the authors described that the indication of pacing included permanent AF for pacemaker implantation. I’m little confusing. The patients with permanent AF only can not be indicated for pacemaker implantation. These patients had permanent AF and high degree AV block? Please clarify.
  3. In page 3, line 121, the authors described that All patients had high-degree atrioventricular conduction disorder, so the authors anticipated high rates of ventricular pacing (>70%). I just wonder if high degree AV conduction disorder means complete AVB? And how did the authors anticipated high rates of ventricular pacing more than 70%? Please add on the evidence or reference.
  4. in page 3, table 1, in RVP group, there were total 10 AF pts and 8 patients had permanent AF with complete AVB and 2 patients had only AF? In HBP, there were total 17 AF pts and 13 patients had permanent AF with complete AVB and 4 patients had only AF?
  5. in page 3, table 1, 14 patients (8 pts in RVP group, 6 pts in HBP group) had coronary artery disease. The authors described that acute coronary syndrome pts were excluded in this study in method section, which means that 14 pts with coronary artery disease were stable angina with/without stent insertion? As you know, some pts with stable angina had critical coronary lesion that needed to do percutaneous coronary intervention, such as stent insertion. And those critical lesions may affect the 2D speckled strain. I think the authors need to add on those data to clarify.
  6. in table 1, total 7 pts (1 pts in RVP group, 6 pts in HBP group) had LBBB/RBBB at the baseline.

I just wonder if there was any difference of speckled strain according to BBB in both group?

  1. the authors showed 1 example of echo assessment using 2D speckle track technique in figure 2.

I just wonder if there are any difference of speckled strain between in patients with complete AV block only vs. in patients with complete AV block + AF? Please add on those data.

Minor

  1. in page 2, the authors described that after the procedure indicated 1D. 1D means 1 day after pacemaker implantation?
  2. in page 5, figure 1, the authors showed example of 12 lead ECG according to pacing and follow-up duration. I’m little confusing. Some ECG were sinus rhythm. 1 ECG was paced QRS. How about showing only ECG with sinus rhythm?
  3. in page 6, in Table 3, the authors showed LAVmax according to pacing type and follow-up duration. Is there any data about LAVI (LAV index)? Because LAVmax is associated with heart volume and size.
  4. how about showing the Table 1, and Table 2 data using graph instead of tables?

Reviewer 2 Report

Congratulations on carrying through a study with excellent design. The data are presented logically and explained in a scientifically sound manner. This topic is trending in this particular field and the authors have done a very good job using electrocardiographic and echocardiographic parameters to show advantages and disadvantages of HBP compared to RVP in regard to ventricular synchrony.

There are no limitations discussed in the study. Clinical outcomes at 1 year+ after the interventions will be a great follow-up to this study.  

Revisions:

Line 44: Change “ventricle” to “ventricular”

Line 58: Change “It is worth of note, that” to “Notably,”

Line 67: Change spelling to “hemodynamic”

Line 69: Change spelling to “hemodynamic”

Line 76: Change “anticipating” to “a need for”

Line 78: Change spelling to “enrollment”

Line 101: Change “echographer” to “sonographer”

Line 116: Remove “n=”

Line 118: Remove “n=”

Line 119: Add after threshold “,respectively”; Remove “n=”

Line 120: Remove “n=”

Line 136: Change “at” to “in”

Line 138: Change “patients” to “pacing (HBP)”

Line 175: Change “at” to “in”

Line 190: Delete “striven to”; Change “demonstrate” to “demonstrated”

Line 197: Delete “usually”

Line 198: Delete “ones”

Line 206: What does “the data” refer to? Be specific or change to “the above data” or “the aforementioned data” if referring to that stated in same paragraph

Line 208: Delete “the most important is that”

Line 209: Insert “risk of” between “the” and “greater”; insert “batter” between “faster” and “consumption”

Line 210: Delete “of the pacemaker”

Line 216: Change first “it” to “QRSd,”; Delete “and it”; Change “triggers” to “triggering”

Line 225: Change “extended” to “increased”

Line 226: Delete “which…duration.”; Change “indicates” to “indicating”

Line 228: Change “shortened” to “decreased”

Line 232: Change “et. el” to “et. al”

Line 233: Change “thesis” to “hypothesis”

Line 238: Chane “that it” to “GLS”

Line 248: Delete “adequately”

Line 249: Delete “The”; insert “development of” in front of “atrial fibrillation”

Line 250: Delete “noticeably”

Line 252: Change spelling to “hemodynamic”

Line 254: Insert “intervention” between “this” and “leads”; insert “clinical” in front of “atrial fibrillation; Delete “in patients treated with cardiac pacing.”

Line 260: Change spelling to “remodeling”; Change “and its” to “that leads to”

Line 261: Change spelling to “hemodynamic”

Figures/Tables:

Table 1: Change “p<” to “p-value”

Table 1: In left column, remove “n/” from all [n/%]

Table 2: Change all “p<” to “p-value”

Table 3: Change all “p<” to “p-value”

Table 3: Try to get data points on single line, that is parts of the standard deviation are displayed on another line

Round 2

Reviewer 1 Report

The authors revised the manuscript well and responsed adequately according to the each comments.